**Data Availability Statement:** All relevant data are within the manuscript files.

**Funding:** The authors received no specific funding for this work.

# Photoinduced electron transfer detection method for identifying *UGT1A1\*28* microsatellites

**Shirou Tsuchida**⊙*, **Noriaki Himi**☯, **Yuuki Miura**☯, **Suzune Kodama**☯, **Tsugumi Shindo**☯, **Koji Nakagawa**‡, **Takashi Aoki**‡

Division of Biochemistry, Department of Molecular Biosciences, School of Pharmaceutical Sciences, Health Sciences University of Hokkaido, Tobetsu-cho, Ishikari-gun, Hokkaido, Japan

☯ These authors contributed equally to this work.
‡ KN and TA also contributed equally to this work.
* tsuchida@hoku-iryo-u.ac.jp

## Abstract

During development of a novel detection method for the *UDP-glucuronosyl transferase 1A1* (*UGT1A1*)*28, the fluorescence intensity of a dye conjugated to cytosine (C) at the end of a DNA strand decreased upon hybridization with guanine (G). This phenomenon is referred to as photoinduced electron transfer (PeT). Using this phenomenon, we devised a method for the naked-eye detection of *UGT1A1\*28* (thymine-adenine (TA)-repeat polymorphism). Fluorescently labeled single-stranded DNA (ssDNA) oligonucleotides (probes) were designed and hybridized with complementary strand DNAs (target DNAs). Base pair formation at the blunt end between fluorescently labeled C (probe side) and G (target side), induced dramatic fluorescence quenching. Additionally, when the labeled-CG pair formed near the TA-repeat sequence, different TA-repeat numbers were discriminated. However, obtaining enough target DNA for this probe by typical polymerase chain reaction (PCR) was difficult. To enable the practical use of the probe, producing sufficient target DNA remains problematic.

## Introduction

The *UDP-glucuronosyl transferase* (*UGT1A1*)*28 is a microsatellite polymorphism consisting of TA-repeats in the *UGT1A1* promoter region. The wild type has six TA-repeats (TA6), while the mutant has seven (TA7) or eight. The mutant expresses less UGT1A1 than the wild type, which is known to induce side effects such as diarrhea in patients treated with irinotecan (Fig 1) [1]. Thus, *UGT1A1\*28* detection is necessary to avoid potential complications [2]. However, the detection by the restriction fragment length polymorphism (RFLP) method [3] is not applicable as, there is currently no available restriction enzyme that directly recognizes and digests the TA-repeat sequence. It is also difficult to detect differences in one of the two bases repeat sequences by allele-specific primer PCR (ASP-PCR) or by the amplification-refractory mutation system (ARMS) method [4]. Currently, the Invader method is the primary *UGT1A1\*28* detection technique [5]. However, this requires a thermal cycler, fluorescence

**Competing interests:** The authors have declared that no competing interest exist.

spectrophotometer, DNA samples, and specialized reagents. Alternative methods include capillary electrophoresis and melting curve analysis, both of which require expensive equipment [6–11]. To overcome these limitations, the "homogeneous assay for fluorescence concentrated on membrane" (HAFCOM), affinity analysis method we developed is applicable to designing a simple and novel genotyping method [12–14]. During the study, it was noticed that for hybridization forming a cytosine-guanine (CG) base pair between a probe possessing a fluorescein isothiocyanate (FITC) labeled terminal cytosine (C) and target DNA with a terminal guanine (G), FITC fluorescence intensity was dramatically reduced. This fluorescence-quenching phenomenon is caused by "photoinduced electron transfer" (PeT) (Fig 2) [15]. Several PeT-based biosensors are already reported [16–18]. The mechanism of PeT in DNA was studied in detail by Mao et al., by comparing the melting curves of fluorophore-labeled ssDNA with complementary ssDNA using real-time PCR. They reported a fluorescence intensity decrease in response to fluorophore-labeled C hybridization with G of the complementary strand at the blunt end [19, 20]. They also explored the effects of temperature and base pairs on fluorophore quenching and demonstrated the usefulness of the base-quenched probe in SNP analysis of the apolipoprotein M (APOM) gene by melting curve [21]. Maruyama T. et al. also attempted to develop SNP genotyping using PeT. They noted a significant decrease in fluorescence intensity at the DNA terminus. The target DNA was hybridized with fluorescently labeled ssDNA (probe), and then restriction enzymes exposed the target SNP at the end of the DNA fragment [22, 23]. Both papers reported a significant decrease in fluorescence intensity upon hybridization between fluorescently labeled C and G at the DNA ends, corroborating the results of our previous experiments. Therefore, we adopted this approach to develop a novel *UGT1A1\*28* detection method able to identify differences in the number of thymine (T) and adenine (A) repeats (microsatellites). The following simpler and less expensive procedure was devised to detect *UGT1A1\*28*: 1) PCR amplification of the microsatellite-containing sequence; 2) combine probe and PCR mixture with heating; 3) probe and target hybridization upon cooling; 4) visual inspection for fluorescence intensity decrease. Advantages of this method are that it is easy to perform on widely available equipment, and is cost-effective as it only requires one

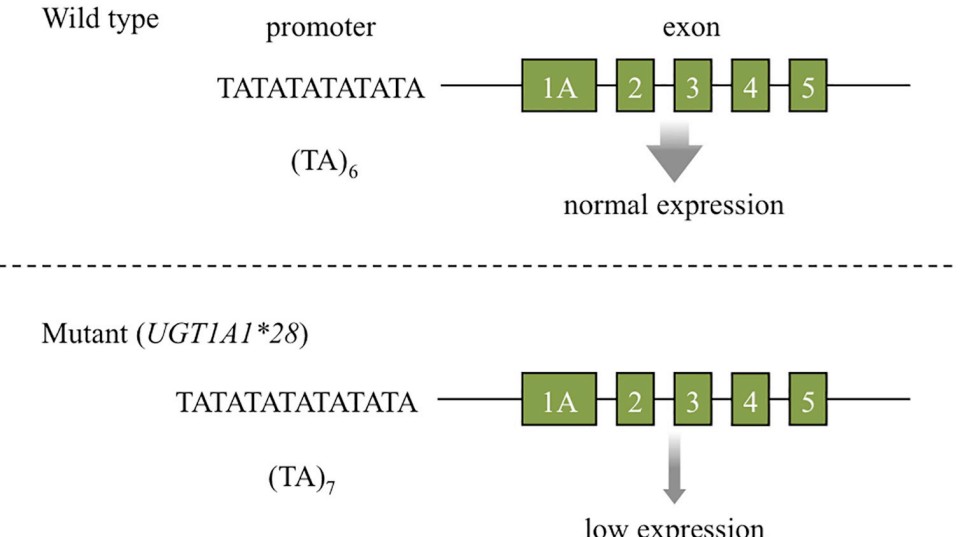

**Fig 1. *UGT1A1\*28*.** Wild type (TA6) promotes greater UGT1A1 expression than the mutant *UGT1A1\*28* (TA7). *UGT1A1\*28* carriers metabolize irinotecan poorly and are prone to side effects (diarrhea).

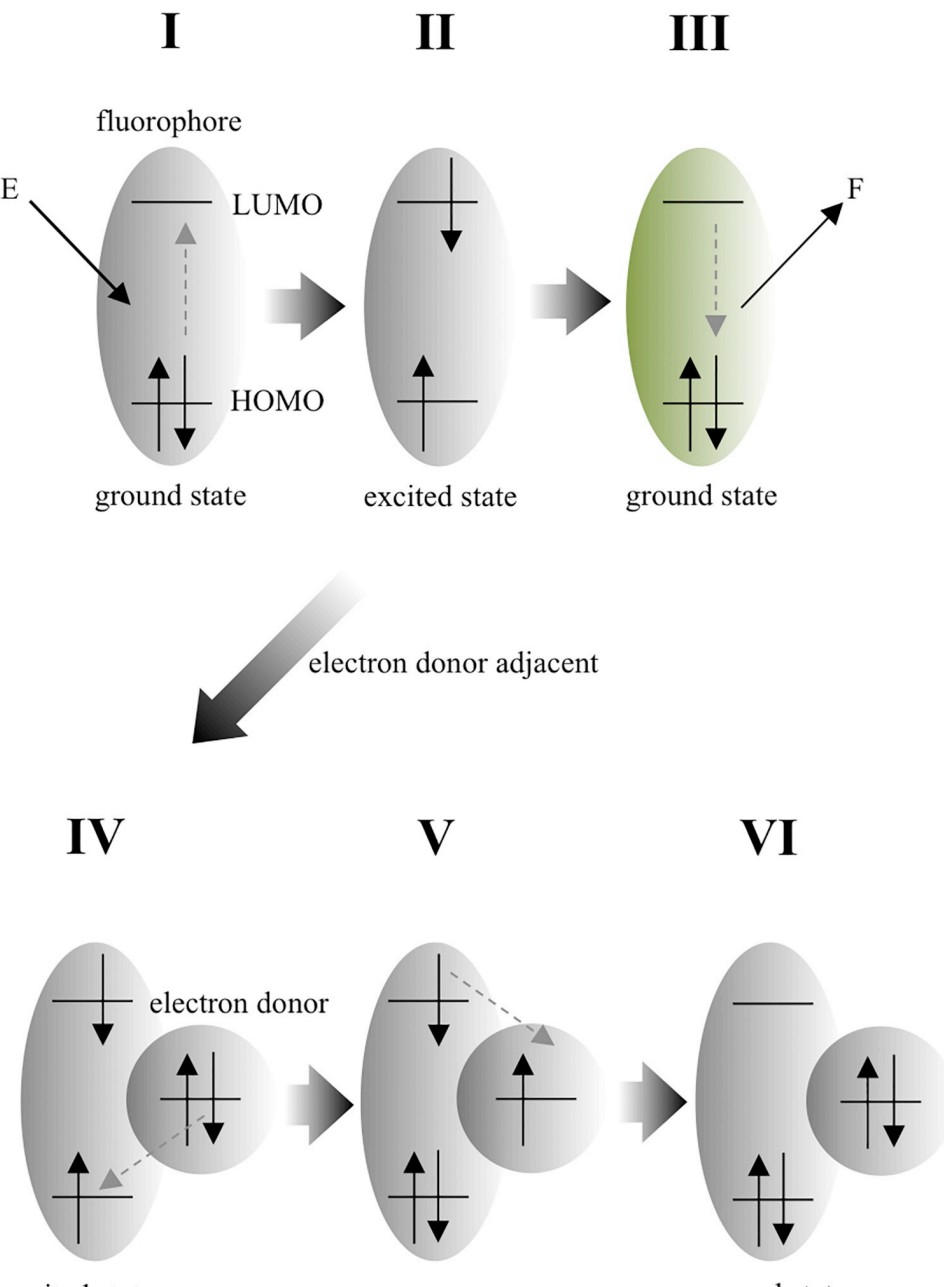

**Fig 2. Principle of PeT.** External energy (E) excites an electron in the highest occupied molecular orbital (HOMO) of the fluorophore to the lowest unoccupied molecular orbital (LUMO) (state I to II). Usually, fluorescence (F) is emitted when an excited electron relaxes to ground state (state II to III). When an electron donor is adjacent to the fluorophore at state II (state II to IV), an electron in HOMO of the electron donor transfers to the lower energy level orbital of the fluorophore (state IV to V). The excited electron of the fluorophore then transfers to the electron donor (here, it is precisely an electron acceptor) (state V to VI). The fluorophore is quenched when the energy levels and proximities of the fluorophore and electron donor are close, enabling electron transfer from the electron donor into the orbital vacancy left by the excited fluorophore electron. The donated electron thus blocks the excited fluorophore electron from relaxing via fluorescence to the ground state, and it transfers to the vacant ground remaining in the electron donor.

fluorescent probe. To realize this, we hybridized probes for target DNA sequences to identify combinations exhibiting the largest fluorescence intensity change. To enable the practical use of this method, sufficient target DNA is required to observe PeT upon hybridization. It is also necessary to obtain target DNA with the TA-repeat sequence as close to the end as possible. Hence, we examined primer sets that amplify the optimal target DNA. As result, we designed a probe that discriminates between TA6 and TA7 repeat numbers, However, PCR conditions that yielded sufficient target DNA for this probe were not established.

## Material and methods

### Chemicals and instruments

Oligonucleotide synthesis (probe, target DNA and primers) was contracted to Thermofisher Scientific (Waltham, MA, USA) and Sigma-Aldrich (St. Louis, MO, USA). Oligonucleotides were dissolved in Tris-EDTA (TE) buffer (10 mM Tris-HCl, 1 mM EDTA, pH 8.0). A DNA polymerase Quick taq HS dyemix (Toyobo, Osaka, Japan) was used for PCR. Agarose was purchased from Nacalai tesque (Kyoto, Japan). Other reagents were purchased from Sigma-Aldrich. A MyCycler thermal cycler (Bio-Rad Laboratories Inc., Hercules, CA, USA) was used in PCR and heating samples. A fluorescence microplate reader MTP-2000FP (Corona Electric, Hitachinaka, Ibaraki, Japan) was used to measure fluorescence intensity. For visual observation of fluorescence, an UV-transilluminator (UVP, Upland, CA, USA) was used and photos were taken using a digital camera. Analytical-grade water was used in all experiments.

### Experimental

**Visual FITC observation.** Various concentrations (0–10.0 μM) of FITC-labeled oligonucleotide (Table 1, probe A) TE solutions in a PCR tubes were irradiated with UV-B (302 nm) on an UV-transilluminator and photos taken using a digital camera.

**Effect of pH on fluorescence intensity.** Various pHs of sodium phosphate buffer (100 mM, pH 3–13) were prepared and 1 μM probe A solutions were prepared with them. Fifty μL of each solution were transferred to a 96-well microplate and fluorescence intensity (ex = 490 nm, em = 530 nm measured using a fluorescence microplate reader.

**Effect of UV irradiation time on fluorescence intensity.** Fifty μL of 1 μM probe A was placed into a 96-well microplate and sealed to prevent drying. Then, the microplate was irradiated with UV-B (302 nm) in a shaded box using an UV-transilluminator. Solution fluorescence intensity (ex = 490 nm, em = 530 nm) was measured at regular intervals (0, 5, 10, 15, 20, 40, 80 min) using a fluorescence microplate reader. Fluorescence intensity at 0 min was taken as 100%.

**Effect of heating on fluorescence intensity.** Sixty μL of 1 μM probe A in a PCR tube was heated at 95°C using a thermal cycler. At regular intervals (0, 15, 30, 60, 120 min), 50 μL of the solution was transferred into a 96-well microplate and fluorescence intensity (ex = 490 nm,

**Table 1. Probe–target combinations for which a decrease in fluorescence intensity (PeT) associated with hybridization was observed.**

| | | Sequence[a] | Length (mer) |
|---|---|---|---|
| Probe A | | 5´–**C**TTA**TATATATATATA₆**TGGCAAAAACCAATCGATACACCA–3´ | 40 |
| Target A | TA6 | 3´–**G**AAT**ATATATATATAT₆**ACCGTTTTTGGTTAGCTATGTGGT–5´ | 40 |
| | TA7 | 3´–GAAT**ATATATATATATAT₇**ACCGTTTTTGGTTAGCTATGTGGT–5´ | 42 |

[a]TA-repeat sequences are underlined. The labeled-cytosine (C) and the guanine (G) that hybridize with the labeled-C are double underlined.

em = 530 nm) was measured using a fluorescence microplate reader. Fluorescence intensity at 0 min was taken as 100%.

**Effect of sodium chloride (NaCl) concentration on fluorescence intensity.** Various concentration (0, 0.5, 1, 2, 4 M) of NaCl solution was prepared with TE buffer. Then 1 μM probe A and target TA6 (Table 1) were prepared with these solutions. Thirty μL of probe A and target TA6 solution was mixed in a PCR tube and heated at 95°C for 2 min using a thermal cycler. Then the tube allowed to cool to room temperature. Fifty μL of mixture was transferred into a 96-well microplate and florescence intensity (ex = 490 nm, em = 530 nm) measured using a fluorescence microplate reader. Fluorescence intensity at 0 M was taken as 100%.

**Effect of protruding end of target DNA.** To investigate the effect of the protruding end on the probe fluorescence intensity (probe B) upon hybridization, targets (target B) shown in Table 2 were designed. The target that forms the blunt end at the 3´ end with 5´ end of the probe was set as the standard (±0G). Targets longer than ±0G are indicated with a plus sign (+), and conversely, targets shorter than ±0G are indicated with a minus sign (−).

**Effect of distance between labeled-CG pair and TA-repeat.** To investigate the effect of the distance between the TA-repeat and the labeled-CG pair on the discrimination of the number of TA-repeats (TA6 or TA7) upon hybridization, the targets (target C) shown in Table 3 were designed. Distance (mer) indicates the number of base pairs between labeled-CG pair and TA-repeat. For example, when probe C 4mer and target C 4mer hybridize, a labeled-CG pair is formed four bases away from the TA-repeat sequence.

**Probe redesign.** To investigate the applicability of the reverse sequence as the target (Fig 3B), additional probes and targets shown in Table 4 were designed. Also, probes shown in Table 5 investigated the impact of florescent labeling internal sequences.

**Probe hybridization to target DNA.** The probe solution (1.0 μM, 30 μL) and target DNA solutions (10.0 μM, 30 μL) were mixed in a PCR tube (final volume was 60 μL). Each final concentrations were 0.5 and 5.0 μM, respectively. Non-complimentary oligonucleotides were used

**Table 2. Probe-target combinations to investigate the effect of protruding ends on fluorescence intensity.**

| | | | Sequence[b] | Length (mer) |
|---|---|---|---|---|
| Probe B | | | 5´-**C**TCCTACTTA**TATATATATATA**$_6$TGGCAAAA-3´ | 30 |
| Target B | +6C[a] | | 3´-GCGGG**AG**AGGATGAAT**ATATATATATAT**$_6$ACCGTTTT-5´ | 36 |
| | +5G[a] | | 3´-CGGG**AG**AGGATGAAT**ATATATATATAT**$_6$ACCGTTTT-5´ | 35 |
| | +4G[a] | | 3´-GGG**AG**AGGATGAAT**ATATATATATAT**$_6$ACCGTTTT-5´ | 34 |
| | +3G[a] | | 3´-GG**AG**AGGATGAAT**ATATATATATAT**$_6$ACCGTTTT-5´ | 33 |
| | +2G[a] | | 3´-GA**G**AGGATGAAT**ATATATATATAT**$_6$ACCGTTTT-5´ | 32 |
| | +1A[a] | | 3´-A**G**AGGATGAAT**ATATATATATAT**$_6$ACCGTTTT-5´ | 31 |
| | ±0G[a] | | 3´-**G**AGGATGAAT**ATATATATATAT**$_6$ACCGTTTT-5´ | 30 |
| | −1A[a] | | 3´-AGGATGAAT**ATATATATATAT**$_6$ACCGTTTT-5´ | 29 |
| | −2G[a] | | 3´-GGATGAAT**ATATATATATAT**$_6$ACCGTTTT-5´ | 28 |
| | −3G[a] | | 3´-GATGAAT**ATATATATATAT**$_6$ACCGTTTT-5´ | 27 |
| | −4A[a] | | 3´-ATGAAT**ATATATATATAT**$_6$ACCGTTTT-5´ | 26 |
| | −5T[a] | | 3´-TGAAT**ATATATATATAT**$_6$ACCGTTTT-5´ | 25 |
| | −6G[a] | | 3´-GAAT**ATATATATATAT**$_6$ACCGTTTT-5´ | 24 |

[a]Plus (+) and minus (−) indicate longer and shorter than probe B, respectively. Numbers (0–6) indicate the number of protruding bases. Bases (A, G, C and T) indicate the base at the 3´ end of target DNA.

[b]TA-repeat sequences are underlined. The labeled-cytosine (C) and the guanines (G) that hybridize with the labeled-C are double underlined.

**Table 3. Probe-target combinations to investigate the effect of distance of FITC from TA-repeat.**

|  | Distance (mer)[a] | TA-repeat | Sequence[b] | Length (mer) |
|---|---|---|---|---|
| Probe C | 4 | 6 | 5´-**C**TTA**TATATATATATA**₆TGGCAAAA-3´ | 24 |
|  | 7 | 6 | 5´-**C**TACTTA**TATATATATATA**₆TGGCAAAA-3´ | 27 |
|  | 8 | 6 | 5´-**C**CTACTTA**TATATATATATA**₆TGGCAAAA-3´ | 28 |
|  | 13 | 6 | 5´-**C**CTCTCCTACTTA**TATATATATATA**₆TGGCAAAA-3´ | 33 |
|  | 24 | 6 | 5´-**C**AGAGGTTCGCCCTCTCCTACTTA**TATATATATATA**₆TGGCAAAA-3´ | 44 |
|  | 25 | 6 | 5´-**C**CAGAGGTTCGCCCTCTCCTACTTA**TATATATATATA**₆TGGCAAAA-3´ | 45 |
| Target C | 4 | 6 | 3´-**G**AAT**ATATATATATAT**₆ACCGTTTT-5´ | 24 |
|  |  | 7 | 3´-GAAT**ATATATATATATAT**₇ACCGTTTT-5´ | 26 |
|  | 7 | 6 | 3´-**G**ATGAAT**ATATATATATAT**₆ACCGTTTT-5´ | 27 |
|  |  | 7 | 3´-GATGAAT**ATATATATATATAT**₇ACCGTTTT-5´ | 29 |
|  | 8 | 6 | 3´-**G**GATGAAT**ATATATATATAT**₆ACCGTTTT-5´ | 28 |
|  |  | 7 | 3´-GGATGAAT**ATATATATATATAT**₇ACCGTTTT-5´ | 30 |
|  | 13 | 6 | 3´-**G**GAGAGGATGAAT**ATATATATATAT**₆ACCGTTTT-5´ | 33 |
|  |  | 7 | 3´-GGAGAGGATGAAT**ATATATATATATAT**₇ACCGTTTT-5´ | 35 |
|  | 24 | 6 | 3´-**G**TCTCCAAGCGGGAGAGGATGAAT**ATATATATATAT**₆ACCGTTTT-5´ | 44 |
|  |  | 7 | 3´-GTCTCCAAGCGGGAGAGGATGAAT**ATATATATATATAT**₇ACCGTTTT-5´ | 46 |
|  | 25 | 6 | 3´-**G**GTCTCCAAGCGGGAGAGGATGAAT**ATATATATATAT**₆ACCGTTTT-5´ | 45 |
|  |  | 7 | 3´-GGTCTCCAAGCGGGAGAGGATGAAT**ATATATATATATAT**₇ACCGTTTT-5´ | 47 |

[a]The distance (mer) indicates the number of bases from the 5´ (probe) or 3´ (target) end of the TA-repeat to where the terminal labeled-CG pair is formed, when the probe and target hybridize.

[b]TA-repeat sequences are underlined. The labeled-cytosines (C) and guanines (G) that hybridize with labeled-C are double-underlined.

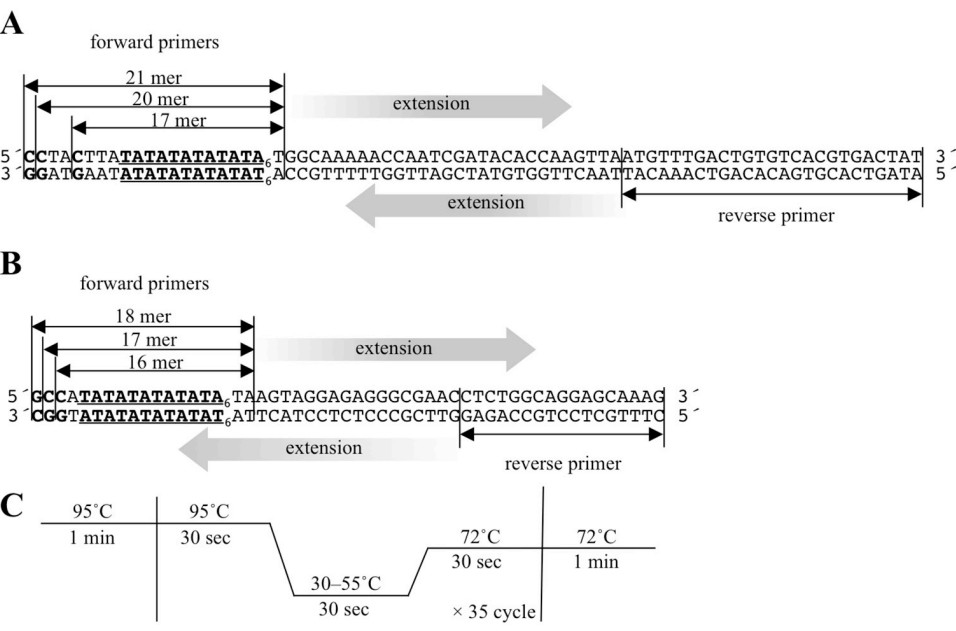

**Fig 3. Trial primer sets for target DNA amplification.** (**A**) Trial primers for target DNA amplification (target C 4mer, 7mer, and 8mer listed in Table 3). (**B**) Trial primers for amplification of target DNAs listed in Tables 4 and 5. (**C**) PCR condition for target amplification.

**Table 4. Redesigned probe-target combinations.**

| | Distance (mr)[a] | Sequence[b] | Length (mer) |
|---|---|---|---|
| Probe D | +3 | 5´-CTTTGCTCCTGCCAGAGGTTCGCCCTCTCCTACTTAT**ATATATATATAT**$_6$GG**C**-3´ | 52 |
| | −1 | 5´-CTTTGCTCCTGCCAGAGGTTCGCCCTCTCCTACTTAT**ATATATATATAT**$_6$GGC-3´ | 52 |
| | −3 | 5´-CTTTGCTCCTGCCAGAGGTTCGCCCTCTCCTACTTAT**ATATATATATAT**$_6$GGC-3´ | 52 |
| Target D | TA6 | 3´-GAAACGAGGACGGTCTCCAAGCGGGAGAGGATGAATA**TATATATATATA**$_6$CCG-5´ | 52 |
| | TA7 | 3´-GAAACGAGGACGGTCTCCAAGCGGGAGAGGATGAATA**TATATATATATATA**$_7$CCG-5´ | 54 |

[a]The distance (mer) from the 5´ end of the TA-repeat. Plus (+) and minus (−) indicate outward and inward directions of the TA repeat sequence, respectively.
[b]The TA-repeat sequences are underlined. FITC-labeled bases are double-underlined.

as controls. The final NaCl concentration was 2.0 M. The mixture was heated at 95°C for 2 min using a thermal cycler and allowed to return to room temperature. Fluorescence intensity was visually inspected with UV-B (302 nm) irradiation of the mixture using an UV-transilluminator and photo taken by digital camera. In addition, the mixture (50 μL) was transferred into a 96-well microplate and the mixture's fluorescence intensity (ex = 490 nm, em = 530 nm) measured using a fluorescence microplate reader. Fluorescence intensity of the probe only solution was taken as 100%.

**Polymerase chain reaction (PCR).** The forward and reverse primers shown in Fig 3A and 3B, were used with our laboratory-conserved plasmid containing a TA6 repeat sequence as the template. Each PCR reaction was performed as follows: heating to 95°C for 1 min; 35 cycles of 95°C for 30 s, 30–55°C for 30 s, and 72°C for 30 s; followed by a final extension for 1 min (Fig 3C). The reaction mixture then undergoes electrophoresis with 1% agarose gel containing ethidium bromide. Amplicon bands were detected using an UV-transilluminato with UV-B (302 nm) and taken photos taken using a CCD camera-equipped imaging system.

## Results and discussion

During our original exploration of molecular affinity analysis, it was observed that upon hybridization forming a CG base pair between a fluorescein isothiocyanate labeled terminal cytosine (C) probe and a target DNA with a terminal guanine (G), FITC fluorescence intensity was dramatically reduced to 40%. However, when the formed GC pair were not at the end, fluorescence intensity only decreased to 80%. This phenomenon was not only observable by the naked-eye, but terminal GC pair formation was identifiable (Fig 4). This fluorescence-quenching phenomenon is caused by PeT. We speculated that this phenomenon could be used to detect mutations without special reagents or equipment. Therefore, we investigate a method to detect *UGT1A1*28* using this phenomenon.

**Table 5. The last probe-target combinations.**

| | | Sequence[a] | Length (mer) |
|---|---|---|---|
| Probe E | | 5´-CTTTGCTCCTGCCAGAGGTTCGCCCTCTCCTACTTAT**ATATATATATAT**$_6$GG-3´ | 51 |
| Target E | TA6 | 3´-GAAACGAGGACGGTCTCCAAGCGGGAGAGGATGAATA**TATATATATATA**$_6$CC-5´ | 51 |
| | TA7 | 3´-GAAACGAGGACGGTCTCCAAGCGGGAGAGGATGAATA**TATATATATATATA**$_6$CC-5´ | 53 |

[a]TA-repeat sequences are underlined. FITC-labeled base is double-underlined.

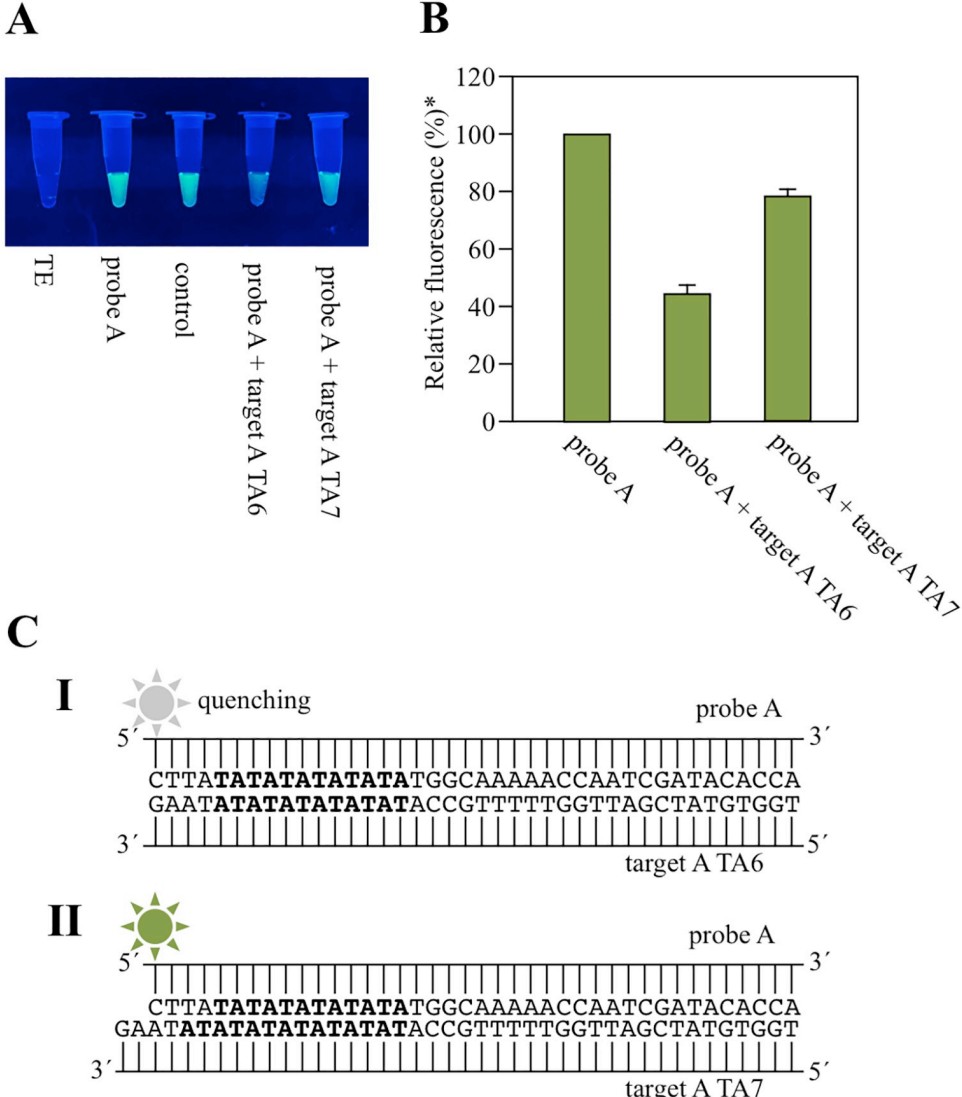

**Fig 4. Discrimination between TA6 and TA7 by PeT.** (**A**) One μM of probe A (30 μL) was added to 10 μM of target A TA6 or TA7 (each 30 μL). The mixtures were then irradiated with UV-B (302 nm) using an UV-transilluminator and observed by the naked-eye. Oligonucleotides non-complementary to the probe were used as controls. (**B**) Fluorescence intensities (ex = 490 nm, em = 530 nm) were measured using a fluorescence microplate reader. The fluorescence intensity of probe A is taken as 100%. (**C**) When FITC-labeled terminal C forms a base pair with the terminal G, fluorescence is quenched (**I**). Conversely, FITC is unquenched when labeled C and G do not form a CG pair at the end (**II**).

## Hybridization conditions

*UGT1A1*28* detection using PeT is indicated by changes in probe fluorescence intensity. FITC (Fig 5A) was chosen as the probe fluorophore for its ubiquitous use. FITC characteristics were carefully examined and dictate the applicable use conditions. First, to confirm the minimum probe concentration at which fluorescence was visually observed, 10 μM probe solution was serially diluted and irradiated with UV-B (302 nm) using an UV-transilluminator, and the fluorescence intensity was visually evaluated. A minimum concentration of roughly 1 μM was required for naked-eye observation of the FITC labeled DNA fluorescence (Fig 5B). Therefore,

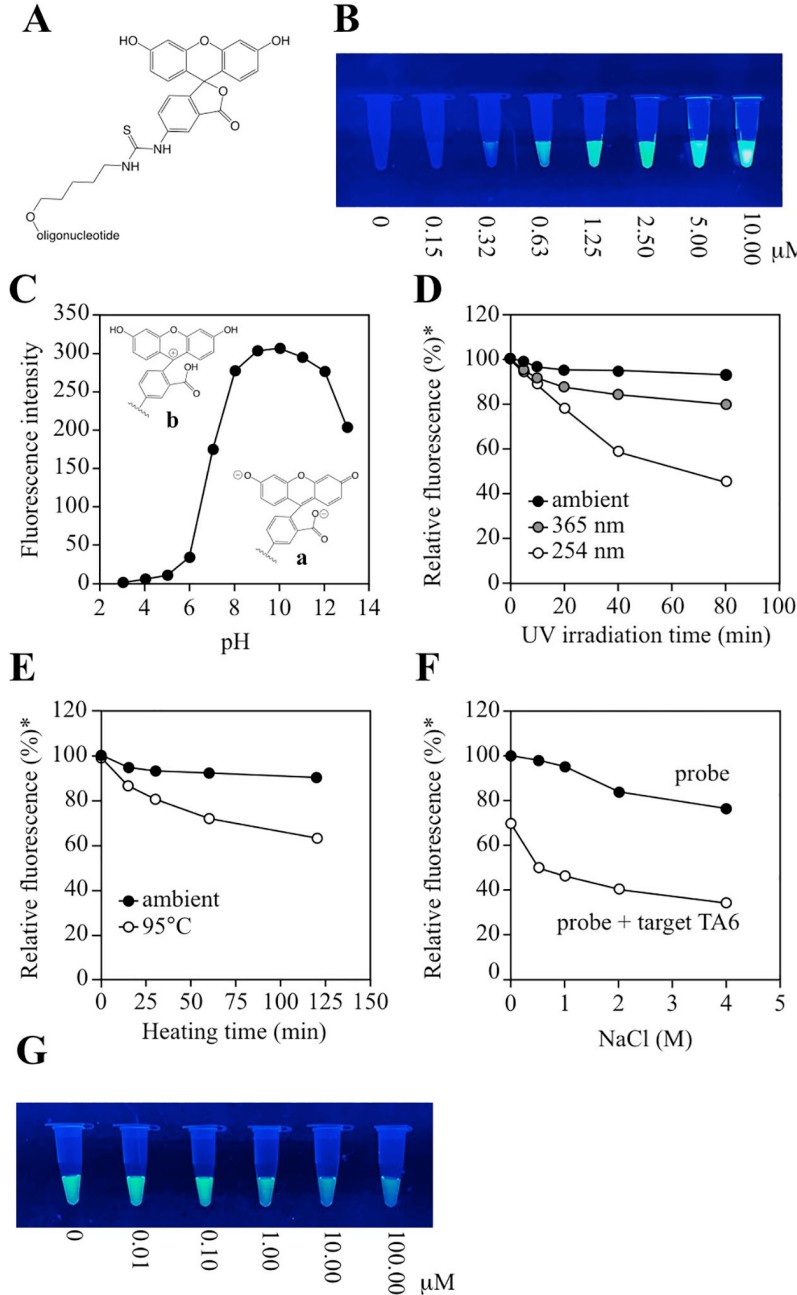

**Fig 5. Characterization of FITC.** (**A**) Structure of FITC labeled oligonucleotide. (**B**) Fluorescence of several concentrations of probe A (Table 1) irradiated by and UV-A observed by the naked-eye using an imaging system. Each volume was 30 μL. (**C**) Effect of pH on fluorescence intensity. Structure **a** and **b** indicate FITC at high and low pH, respectively. (**D**) Influence of UV irradiation time on fluorescence intensity. *The fluorescence intensity at 0 min is taken as 100%. (**E**) Influence of heating time on fluorescence intensity. *The fluorescence intensity at 0 min is taken as 100%. (**F**) Influence of NaCl concentration on fluorescence intensity and hybridization. *The fluorescence intensity at 0 M NaCl is taken as 100%. (**G**) One μM probe A was mixed with various concentration of target A TA6 (0–100 μM).

the probe concentration was set to 1 μM for subsequent experiments. Then, change in the fluorescence intensity of FITC in response to pH was scrutinized. The probe fluorescence intensity was measured at various pHs, with the fluorescence intensity decreasing dramatically under acidic conditions (Fig 5C). Under neutral to alkaline condition FITC forms the di-anionic form (Fig 5C, a), absorbing blue light (around 490 nm) and emitting green light (around 520 nm). In contrast, under acidic conditions, FITC becomes either neutral (Fig 5C, a) or cationic (Fig 5C, b). Both absorb significantly less blue light and do not fluoresce [24–26]. Typical nucleic acid experiments buffers are TAE (40 mM Tri-acetate, pH 8.3, 1 mM EDTA), TBE (89 mM Tris-borate, pH 8.3, 2 mM EDTA) and TE (10 mM Tris-HCl, pH 8.0, 1 mM EDTA). These typically (around pH 8.0) did not obviously affect FITC fluorescence intensity. Thus, TE buffer was used in subsequent experiments. Next, the influence of UV irradiation time upon FITC fluorescence intensity was investigated as UV irradiation is necessary to observe fluorescence. In a light-shielded box, the probe solution was irradiated with UV light using an UV-transilluminator, FITC fluorescence diminished to 90% under UV irradiation after about 10 min (Fig 5D). This is within tolerances, as *UGT1A1\*28* detection by UV irradiation is expected to be performed within a few minutes. For *UGT1A1\*28* detection, it is assumed that the probe is added to the target DNA (amplicon) solution, the mixture is heated once to 95°C to denature the double-strand DNA, followed by probe hybridization with the target DNA upon returning to room temperature. Accordingly, the effect of heating on probe florescence intensity was investigated. Although FITC fluorescence diminished to 90% upon heating at 95°C for about 10 min (Fig 5E), the thermal stability of FITC is also sufficient as the detection heating time is approximately about 2 min. As NaCl concentration generally enhances DNA hybridization, the effect of NaCl concentration on fluorescence intensity changes produced by probe-target hybridization was further examined. Concentrated NaCl (2.0–4.0 M) resulted in a fluorescence intensity decrease of about 20%. At high NaCl concentration (>2M), FITC fluorescence decreased to about 80% (Fig 5F). The cause of this phenomenon was unclear. However, subsequent experiments were performed in the presence of 2.0 M NaCl in TE buffer (pH 8.0) as the NaCl addition enhanced probe and target hybridization and facilitated fluorescence attenuation (Fig 5D). Although 2M NaCl is considered to induce nonspecific hybridization in practical use, robust hybridization of probe and target was prioritized for this fundamental study. The impact of NaCl concentration on nonspecific hybridization will be investigated in a practical application study. High probe concentrations against target DNA result in large background fluorescence noise. In contrast, excess target DNA concentration against the probe did not induce a reduction in fluorescence intensity (Fig 5G). In this study, it was assumed that as a large amount of target DNA may be obtained by PCR, the concentration ratio of probe to target DNA was set at 1:10.

## Probe design

To investigate the protruding end on PeT, various target DNAs were designed as listed in Table 2. The probe-target combinations in Table 2 were hybridized and the fluorescence intensity evaluated. The greatest FITC fluorescence intensity decrease occurred when the terminal CG pair formed a blunt end (±0G in Fig 6A). A fluorescence intensity decrease was also observed when the probe side formed a protruding end, although this effect was less pronounced than that for the blunt end (+6C–+1A in Fig 6A). When the target side was the protruding end, the fluorescence intensity decreased as the length of the protruding end became shorter (−2G––6C in Fig 6A). However, one case deviated from this trend, as observed in target B –1A. Therefore, the fluorescence intensity decrease is expected to be influenced not only by the blunt or protruding ends, but also by the adjacent bases. Tentatively, it was

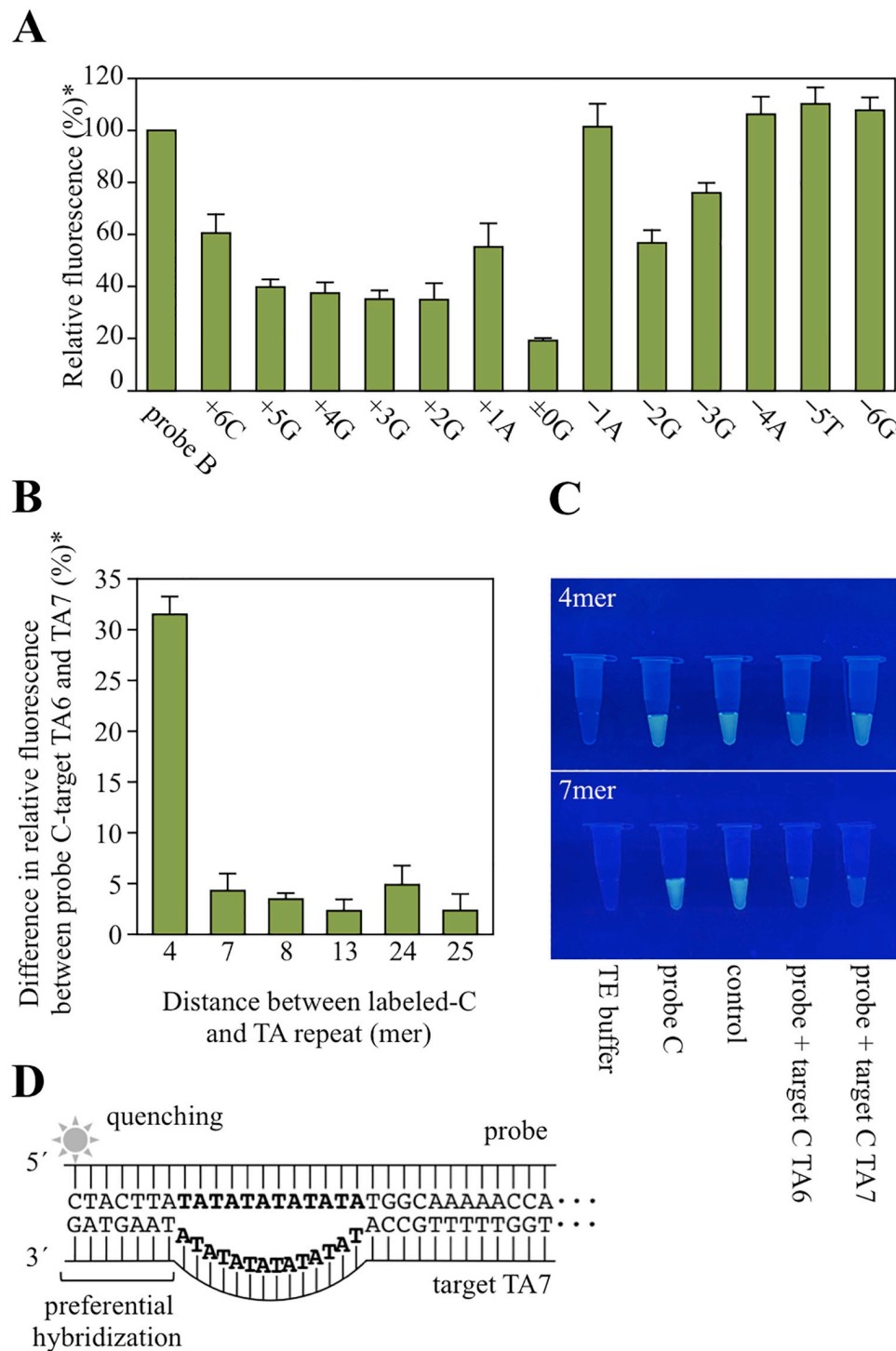

**Fig 6. Impact of surrounding bases on FITC labeled C and G pair fluorescence intensity.** (**A**) Influence of protruding ends on fluorescence intensity. Probe and targets are listed in Table 2. *The fluorescence intensity of probe B is taken as 100%. (**B**) Effect of distance from labeled-CG pair to TA-repeat sequence on fluorescence intensity. Probe and targets are listed in Table 3. *The difference in fluorescence intensity between probe C and TE buffer is taken as 100%. (**C**) Naked-eye images of probe-target mixture. Above photo shows probe C 4mer and target C 4mer mixture. Below photo shows probe C 7mer and target C 7mer mixture. (**D**) Hypothesis for the lack of fluorescence intensity decrease upon probe hybridization to the TA7 target.

decided to design a probe in which the labeled-C (probe side) and G (target side) form base pair at the blunt end, at which the fluorescence intensity decreased the most. In the human UGT1A1 promoter, the closest G to the TA-repeat is four bases away from the TA-repeat. The next closest G is 7 bases away, the next 8 bases away, and the next is 13 bases away, and so on. Thus, six different targets with G as the 3′ end at various distances from TA-repeat (up to 24 bases away) were designed. Then, probes in which the labeled-C forms base pairs with these targets at the blunt end were designed (Table 3). Fluorescence intensity changes due to the distance from TA-repeat sequences to labeled-CG pairs were examined for the probe-target combinations listed in Table 3. For the probe C 4mer and target C 4mer (4 bases away from the TA-repeat sequence), the greatest difference in fluorescence intensity was observed between TA6 and TA7, and was easily visually distinguished. However, the combination of probe C 7mer and target C 7mer produced only minor fluorescence intensity changes between TA6 and TA7 (Fig 6B and 6C). Similarly, TA6 and TA7 discrimination was not achieved for the combination in which the labeled C and G formed base pairs more than seven bases away from the TA-repeat sequence. These results indicate that to increase the fluorescence intensity difference between the target TA6 and TA7, the probe-side fluorescence label must be as close as possible to the TA-repeat sequence. That PeT was also observed in the combination of probe and target TA7 may be due to TA-repeat sequence loop-out. The number of hydrogen bonds between adenine and thymine (two bonds) is less than that between guanine and cytosine (three bonds), suggesting that TA-repeat sequence is less amenable to hybridization than other sequences. In other words, the further away the labeled-CG pair is from the TA-repeat sequence, the more strongly the other sequence is expected to hybridize with looping-out of the TA-repeat (Fig 6D). Based on these results, the following three factors are considered important in designing the PeT probe for *UGT1A1*28*. First, the C labeled with a fluorophore (probe side) and G (target side) must hybridize. Second, the labeled C and G must hybridize at the end of DNA. And third, the labeled C must be as close to the TA-repeat sequences as possible.

## Amplification of the target DNA

To enable the practical use of this method, target DNA must be obtainable by PCR using the subject's genomic DNA as a template. As mentioned above, when the probe and target hybridize, the labeled-CG pair must form at the end and as close to the TA-repeat as possible. Therefore, primers to amplify the target required designing such that they overlap the TA-repeat sequence. If the target is amplified with primers straddling the TA-repeat sequence, the original TA-repeat count becomes replaced by the primer-derived TA-repeat count. This reflects principle of the site-directed mutagenesis method. Moreover, the 5′ end of primer must be C and as close to the TA-repeat as possible. Excessively short primers, although inappropriate for PCR, were necessary to obtain target that induced PeT. Thus, we examined the PCR conditions (especially annealing temperature) using the primer sets shown in Fig 3A. At low annealing temperatures (<45˚C), the target DNA may be amplified using forward primer 21mer or forward primer 20mer (Fig 7A, see 21mer and 20mer). The amplicon obtained with forward primer 21 and 20mer correspond to target C TA6 8mer and 7mer in Table 3, respectively. However, the above results (Fig 6B and 6C) suggest that TA6 and TA7 amplicon discrimination is difficult. When the amplicon obtained with primer 17mer, shown in Fig 3A, was used as target DNA, it was expected to discriminate between TA6 and TA7, but amplification did not occur, even when the annealing temperature was set to 30˚C (the lowest thermal cycler temperature available) (Fig 7A, see 17mer). Hence, new primer sets were designed with the aim of generating PeT at the CG pair of the opposite TA-repeat sequence (Fig 3B). Similarly,

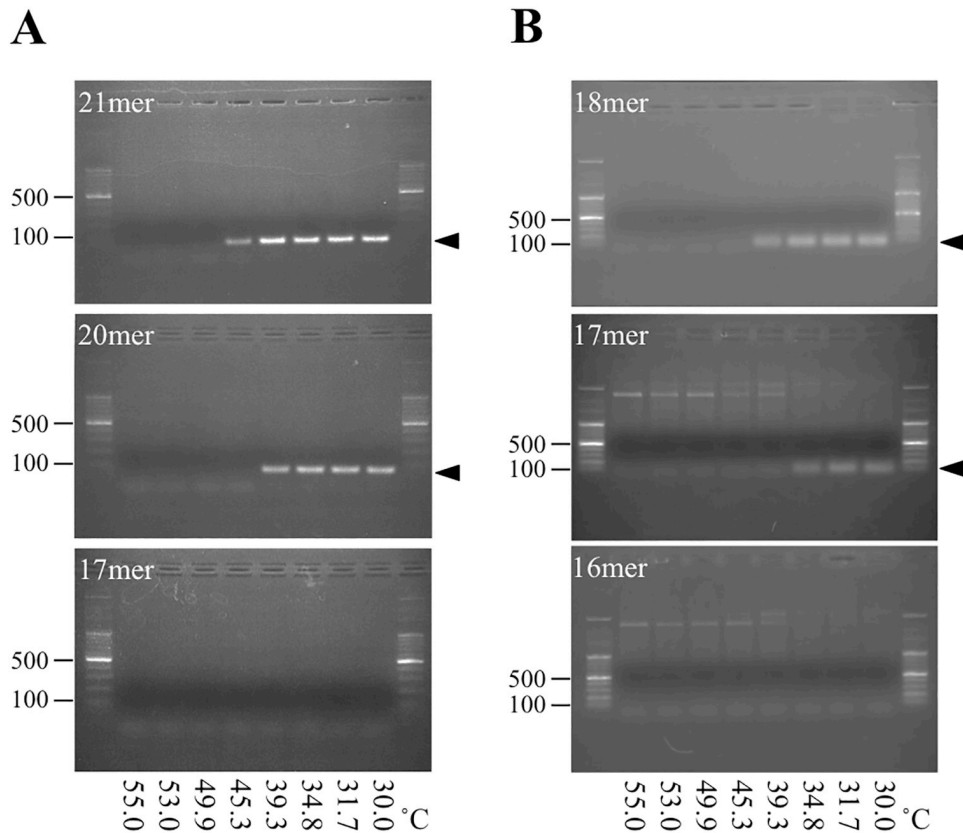

**Fig 7. Amplification of target DNA at different annealing temperatures.** (**A**) Agarose gel electrophoresis images of PCR mixtures at various annealing temperatures (30–55˚C) using the primer sets in Fig 3A are shown. (**B**) Similarly, the primer sets used in Fig 3B are shown. The target DNA bands are indicated by closed arrows.

we attempted to amplify target DNA at various annealing temperatures and were able to amplify a small amount of target DNA that resulted in PeT close to the TA-repeat sequence at a temperature lower than 35˚ C (Fig 3B, see 18mer and 17mer). If primers with high TA content are designed, a lower annealing temperature may be used. In general, low annealing temperature induces nonspecific amplification. This issue will be examined in a future practical application study.

## Probe redesign

In response, the probes were redesigned to match the target DNA expected when using an 18mer or 17mer primer (Tables 4 and 5). Here, we wondered if loop-out may be used to detect *UGT1A1*28*. In redesigning the probe, we examined whether the fluorescence label added to the probe internal sequence is quenched by PeT. In other words, while the formation of CG pair leads to PeT significantly decreasing fluorescence intensity, non-formation of CG pairs and loop-out synergistically suppress decreases in fluorescence intensity. Thus resulting in a grater fluorescence intensity difference between TA6 and TA7. For labeling of bases inside the sequence, Thermofisher Scientific, oligo DNA supplier, was contacted and were informed that thymine (T) may label even inside the sequence. Therefore, for probes D –1mer, probe D – 3mer, and probe E, the fluorescent label was conjugated to the thymine (T) inside the sequence instead of the cytosine (C) at the end. This avoids the CG pair formation associated with loop-

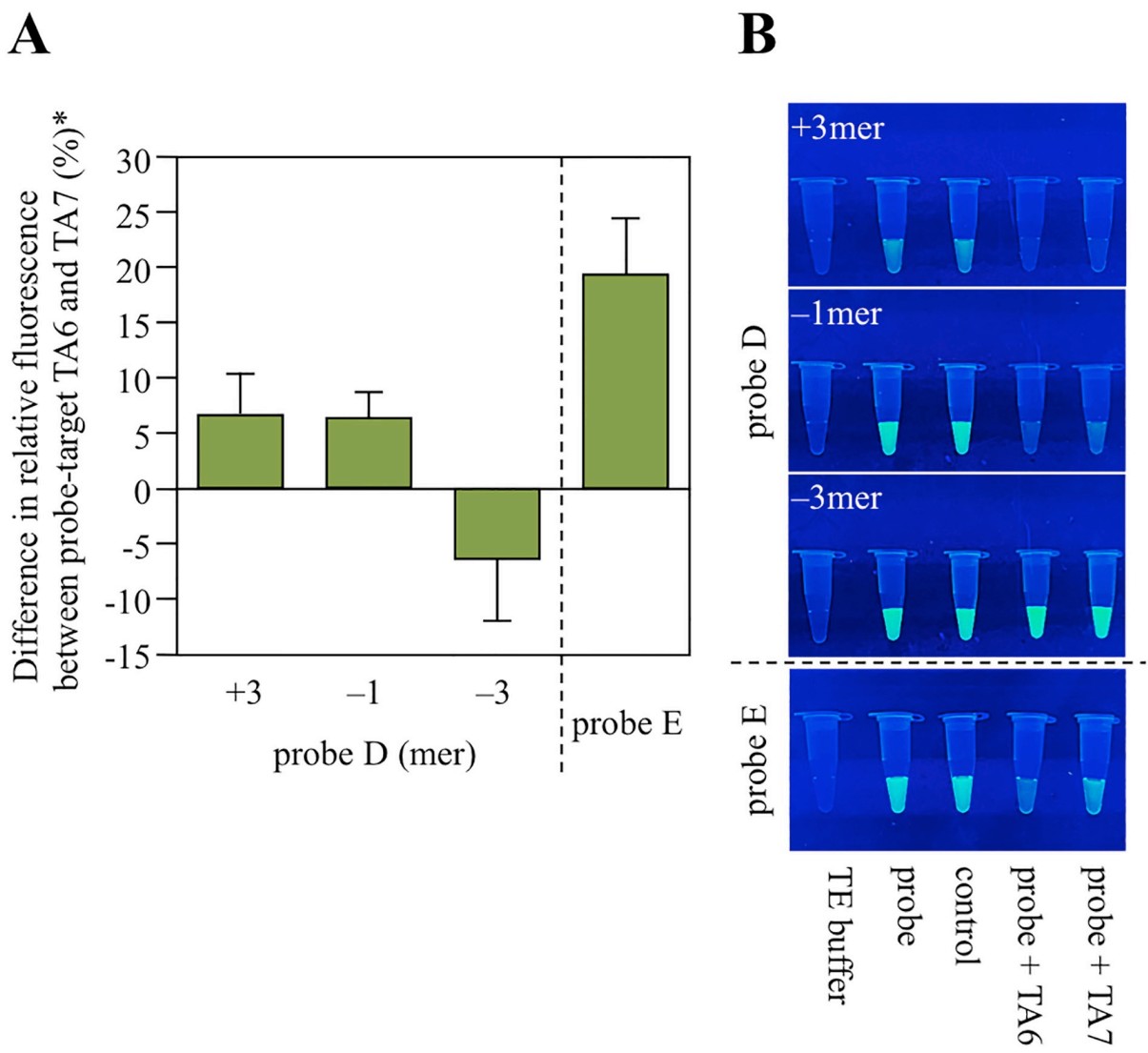

**Fig 8. Fluorescence intensity of redesigned probes and target DNA combination.** (**A**) Fluorescence intensity differences between probe-target mixtures listed in Tables 4 and 5. *Probe fluorescence intensity is taken as 100%. (**B**) Naked-eye image of probe-target mixtures listed in Tables 4 and 5.

out of the TA-repeat sequence hypothesized in Fig 6D, thus enhancing the discrimination between TA6 and TA7. Hybridization of these redesigned probes with their targets resulting in TA 6 and TA7 not clearly identifiable with probe D, but visually discriminated with probe E (Fig 8). As PeT occurs when an electron donor and fluorophore are in proximity, a fluorescence decrease was observed by the naked-eye, even for the probe modified within the sequence, somewhat contrary to the constraints noted in the previous section. *UGT1A1*28* is detected by probe E addition to the PCR reaction solution using the primer set (Fig 3B, forward primer 17mer and reverse primer), heating, cooling, and measuring (or visually observing) the fluorescence intensity. However, this method requires a 10-fold excess of target DNA to probe. The minimum probe concentration to visually confirm fluorescence is approximately 1 µM, and the target DNA must be roughly 10-fold excess (about 10 µM). As the amplicon concentration of 50 base pairs obtained by typical PCR (when the primer concentration is

10 μM) is ≤ 0.2 μM [27], it is necessary to either increase the amplicon 10-fold or to concentrate the PCR reaction solution 10-fold. PCR method improvements that significantly increase the amplicon concentrations may enable the practical use of this method [28, 29]. Particularly, target DNA amplification as ssDNA is best [30]. In addition, the *UGT1A1*28* detection method may be applied to discriminate other microsatellite polymorphisms with minor differences in repeat numbers. The larger the number of repeats, the larger the expected difference in fluorescence intensity between genotypes. In this study, we focused on discrimination of TA6/TA6 homozygous and TA7/TA7 homozygous. However, heterozygous TA6/TA7 does exist. Discrimination between homozygous and heterozygous forms is a focus for future research.

## Conclusions

We attempted to develop a PeT-based *UGT1A1*28* detection method. When a fluorescence-labeled probe hybridized with target DNA, difference in fluorescence intensity was observed between the wild-type and mutant-type the naked-eye. This method did not require special and expensive equipment. The problem in clinical application of this method is how to obtain the desired target DNA in large quantities.

## Supporting information

**S1 Raw images. These photos used in Fig 7 (amplification of target DNA at different annealing temperatures).** (**A**) Agarose gel electrophoresis images of PCR mixtures at various annealing temperatures (30–55˚C) using the forward primer 17 mer and reverse primer in Fig 3A. (**B**) Similarly, the forward primer 20 mer and reverse primer in Fig 3A. (**C**) Similarly, the forward primer 21 mer and reverse primer in Fig 3A. (**D**) Similarly, the forward primer 18 mer and reverse primer in Fig 3B. (**E**) Similarly, the forward primer 17 mer and reverse primer in Fig 3B. (**F**) Similarly, the forward primer 16 mer and reverse primer in Fig 3B. All photos were taken with a CCD camera attached to BioDoc-It imaging system (UVP, Upland, CA, USA) under UV-B (302 nm) irradiation. Photo A, B and C were printed out without saving the digital data, therefore photos were scanned and edited. All photos were cropped appropriate. Photo A, B and C were flipped horizontally.
(PDF)

## Author Contributions

**Conceptualization:** Shirou Tsuchida.

**Investigation:** Shirou Tsuchida, Noriaki Himi, Yuuki Miura, Suzune Kodama, Tsugumi Shindo.

**Methodology:** Shirou Tsuchida.

**Project administration:** Shirou Tsuchida.

**Supervision:** Koji Nakagawa, Takashi Aoki.

**Validation:** Shirou Tsuchida.

**Visualization:** Shirou Tsuchida.

**Writing – original draft:** Shirou Tsuchida.

**Writing – review & editing:** Shirou Tsuchida.

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
