## [Decision Letter · Decision Letter 0]

7 Jul 2023

PONE-D-23-16592Photoinduced electron transfer detection method for identifying UGT1A1*28 microsatellitesPLOS ONE

Dear Dr. Tsuchida,

Thank you for submitting your manuscript to PLOS ONE. After careful consideration, we feel that it has merit but does not fully meet PLOS ONE’s publication criteria as it currently stands. Therefore, we invite you to submit a revised version of the manuscript that addresses the points raised during the review process.

We look forward to receiving your revised manuscript.

Kind regards,

Hari S. Misra, Ph.D.

Academic Editor

PLOS ONE

Journal Requirements:

Reviewers' comments:

Reviewer's Responses to Questions

**Comments to the Author**

1. Is the manuscript technically sound, and do the data support the conclusions?

Reviewer #1: Yes

Reviewer #2: Yes

2. Has the statistical analysis been performed appropriately and rigorously? 

Reviewer #1: Yes

Reviewer #2: Yes

3. Have the authors made all data underlying the findings in their manuscript fully available?

Reviewer #1: Yes

Reviewer #2: Yes

4. Is the manuscript presented in an intelligible fashion and written in standard English?

Reviewer #1: Yes

Reviewer #2: Yes

5. Review Comments to the Author

Reviewer #1: The manuscript titled Photoinduced electron transfer detection method for identifying UGT1A1*28 microsatellites reports on the development of a novel detection method for the UDP-glucuronosyl transferase 1A1 (UGT1A1)*28. The authors have employed the fluorescence intensity of a dye conjugated to cytosine (C) at the end of a DNA strand decreased upon hybridization with guanine (G) and this process is known as the photoinduced electron transfer (PeT). Using this phenomenon, the authors also devised a method for the naked-eye detection of UGT1A1*28 (thymine-adenine (TA)-repeat polymorphism). Fluorescently labeled single-stranded DNA (ssDNA) as oligonucleotides (probes) were designed and hybridized with complementary strand DNAs; c-DNA (as target DNAs). Base pair formation between fluorescently labeled C (probe side) and G (target side) induced dramatic fluorescence quenching. In addition, when the labeled-CG pair formed near the TA-repeat sequence, different TA-repeat numbers were discriminated. Despite some success with the proposed novel study, obtaining sufficient amount of target DNA for this probe by polymerase chain reaction (PCR) was difficult to enable practical use of probe.

The manuscript is reasonably well-written except some minor changes that are required.

The Fig. 5 needs attention and occasionally the edits throughout the manuscript. Once these comments are taken cared, the manuscript is acceptable for publication in this journal.

Reviewer #2: Shirou Tsuchida et al. proposed a new method for DNA complementary detection of UGT1A1*28 microsatellites based on photoinduced electron transfer. This method shows certain analytical ability for base mutations and has certain practical value. It is recommended that the paper be accepted after the author has addressed the following questions.

There are some minor problems:

1. The author claims that naked-eye inspection can be performed. What does it mean exactly?

2. Is the detection method proposed by the author specific?

3. What are the advantages of this method compared with other existing methods?

4. The format of the references is incomplete, such as references 11 and 26. It is recommended to check carefully.

6. PLOS authors have the option to publish the peer review history of their article (what does this mean?). If published, this will include your full peer review and any attached files.

Reviewer #1: No

Reviewer #2: No

---

## [Author Response · Author response to Decision Letter 0]

13 Jul 2023

Response to the Reviewer #1

Thank you for your review of our paper. We have answered your points below. 

The Fig. 5 needs attention and occasionally the edits throughout the manuscript. Once these comments are taken cared, the manuscript is acceptable for publication in this journal.

Answer: We thank for your helpful suggestion. We have checked all figures and corrected any misalignment of objects and text in the figures. We have also changed all figures from doc format to tiff format according to PLOS ONE guideline. We have checked all figures. Other changes to the manuscript are shown red bold in the revised manuscript.

Once again, thank you for your review and helpful comments.

Response to the Reviewer #2

Thank you for your review of our paper. We have answered each your points below. Changes to the manuscript are shown blue bold in the revised manuscript.

1. The author claims that naked-eye inspection can be performed. What does it mean exactly?

Answer: Thank you for your comment. When a florescence-labeled probe is hybridized to a target single-strand DNA, the fluorescence intensity of the probe decreases when the target DNA has six TA repeats, while the fluorescence intensity of the probe does not decrease when the target DNA has seven TA repeats. This difference in florescence intensity can be observed with the naked-eye. In other words, we can detect the difference in number of TA repeats (UGT1A1*28) as the difference in fluorescence intensity with the naked-eye. We have shown this result in Fig 8E.

2. Is the detection method proposed by the author specific?

Answer: Thank you for your comment. We are very sorry. We are not sure what "specific" mean. If you mean "specific = original", then the answer is "yes"; We are the first to apply it to UGT1A1*28 detection, although PeT is a well-known phenomenon. If you mean "specific = limited", the answer is "yes". The probe used in this study is limited to UGT1A1*28 detection. It can be applied to other microsatellite detection by changing the probe. Alternatively, if "specific = concrete", the answer is "No". We have not applied this method on human. The concern here is that PCR using genomic DNA collected from the subject will not yield a sufficient amount of target DNA. If "specific = precise", answer is "no". Even though this method can identify homozygotes (TA6/TA6 and TA7/TA7). We need to design a better probe for identification of homozygous and heterozygous.

3. What are the advantages of this method compared with other existing methods?

Answer: The advantage of this method is that dose not require special (and expensive) equipment or reagents. While other methods require both a thermal cycler and a fluorescent microplate reader, or a real-time PCR device, this method requires only a standard thermal cycler and UV light. We have added this point to the conclusion section (line 510–511). 

4. The format of the references is incomplete, such as references 11 and 26. It is recommended to check carefully.

Answer: We have corrected reference 11 and 26 in REFERENCES section (line 557 and 606). We have also checked and corrected other references, according to the PLOS ONE guidelines.

Once again, thank you for your review and helpful comments.

---

## [Decision Letter · Decision Letter 1]

20 Jul 2023

Photoinduced electron transfer detection method for identifying UGT1A1*28 microsatellites

PONE-D-23-16592R1

Dear Dr. Tsuchida,

We’re pleased to inform you that your manuscript has been judged scientifically suitable for publication and will be formally accepted for publication once it meets all outstanding technical requirements.

Kind regards,

Stephen D. Ginsberg, Ph.D.

Section Editor

PLOS ONE

Reviewers' comments:

Reviewer's Responses to Questions

**Comments to the Author**

1. If the authors have adequately addressed your comments raised in a previous round of review and you feel that this manuscript is now acceptable for publication, you may indicate that here to bypass the “Comments to the Author” section, enter your conflict of interest statement in the “Confidential to Editor” section, and submit your "Accept" recommendation.

Reviewer #2: All comments have been addressed

2. Is the manuscript technically sound, and do the data support the conclusions?

Reviewer #2: Yes

3. Has the statistical analysis been performed appropriately and rigorously? 

Reviewer #2: Yes

4. Have the authors made all data underlying the findings in their manuscript fully available?

Reviewer #2: Yes

5. Is the manuscript presented in an intelligible fashion and written in standard English?

Reviewer #2: Yes

6. Review Comments to the Author

Reviewer #2: Many thanks to the author for the response. I think the manuscript can be accepted for publication in its current form

7. PLOS authors have the option to publish the peer review history of their article (what does this mean?). If published, this will include your full peer review and any attached files.

Reviewer #2: No

---

## [Editor Report · Acceptance letter]

25 Jul 2023

PONE-D-23-16592R1 

Photoinduced electron transfer detection method for identifying UGT1A1*28 microsatellites 

Dear Dr. Tsuchida:

I'm pleased to inform you that your manuscript has been deemed suitable for publication in PLOS ONE. Congratulations! Your manuscript is now with our production department. 

Kind regards, 

on behalf of

Dr. Stephen D. Ginsberg 

Section Editor

PLOS ONE